# Progressive Damage Numerical Modelling and Simulation of Aircraft Composite Bolted Joints Bearing Response

**DOI:** 10.3390/ma13245606

**Published:** 2020-12-08

**Authors:** Guoqiang Gao, Luling An, Ioannis K. Giannopoulos, Ning Han, Ende Ge, Geng Hu

**Affiliations:** 1Jiangsu Key Laboratory of Precision and Micro-Manufacturing Technology, Nanjing University of Aeronautics and Astronautics, Nanjing 210016, China; hanning@nuaa.edu.cn; 2Centre of Aeronautics, School of Aerospace, Transport and Manufacturing, Cranfield University, Cranfield MK43 0AL, UK; 3Institute of Aeronautical Manufacturing Technology, Shanghai Aircraft Manufacturing Co. Ltd., Shanghai 200126, China; geende@comac.cc; 4AECC Hunan Aviation Powerplant Research Institute, Zhuzhou 412002, China; Micron1012@163.com

**Keywords:** structural joints, mechanical testing, strength, numerical modelling

## Abstract

Finite element numerical progressive damage modelling and simulations applied to the strength prediction of airframe bolted joints on composite laminates can lead to shorter and more efficient product cycles in terms of design, analysis and certification, while benefiting the economic manufacturing of composite structures. In the study herein, experimental bolted joint bearing tests were carried out to study the strength and failure modes of fastened composite plates under static tensile loads. The experimental results were subsequently benchmarked against various progressive damage numerical modelling simulations where the effects of different failure criteria, damage variables and subroutines were considered. Evidence was produced that indicated that both the accuracy of the simulation results and the speed of calculation were affected by the choice of user input and numerical scheme.

## 1. Introduction

The use of resin-based carbon fiber composite materials in modern large civil aircraft has been increasing significantly in recent decades. Bolted joints are still widely used on composite airframe structures due to their ease of installation and disassembly and their damage tolerance characteristics. The strength estimation in terms of bolt bearing against composite laminated plates is important for the design optimization of aircraft composite structures. Over the past 30 years, a significant amount of research has been published regarding the numerical and experimental behaviors of composite bolted joints.

Analytical models have been widely used to predict the behavior of single-lap composite bolted joints, and the effects of geometrical parameters, material properties, the stacking sequences of composites, bolt torque and the friction coefficient on the stiffness of composite bolted joints have been included in these analytical models. These models are regarded as valuable preliminary tools for the analysis of the stiffness of composite bolted joints [1,2,3,4,5].

Finite element technology has been applied in the current study to determine the structural behaviors of composite bolted joints where not only could strain and stress be calculated for the whole loading process, but different composite damage states could be simulated. The early three-dimensional progressive damage models used to analyze composite joint series mainly incorporated Hashin [6] (or maximum-stress failure criteria), Chang-Chang [7] or Tan degradation [8], and the accuracy of these simulations were mainly evaluated according to the numerical values of the strength and stiffness of composite joints [9,10]. The effects of failure criteria and degradation rules on the prediction results were then valued, and it was found that strength was sensitive to the selected and degradation factors [11,12].

Álvaro Olmedo and Carlos Santiuste proposed a set of failure criteria based on Chang–Lessard criteria [13], considering the effect of out-of-plane stress and a non-linear shear stress–strain relationship [14]. The load-displacement curve predicted by the new failure criteria was closer to the experimental result than the curve predicted using the Hashin criteria. C. Hühne and A.-K. Zerbst et al. applied Hashin criteria with constant and continuous degradation models to determine the progressive damage of composite bolted joints with liquid shim layers, wherein the material data of the continuous degradation model was compiled by a MATLAB routine and implemented in finite element software ABAQUS at same time. The numerical results acquired by continuous degradation showed better correlation with the experimental data [15].

The abovementioned progressive damage models for carbon-fiber composite joints available in the literature were mainly based on the ABAQUS user subroutine USDFLD (User subroutine to redefine field variables at a material point). Although different failure criteria can be conveniently applied by USDFLD, only constant degradation rules can be used conveniently. A complex constitutive model such as an elastic–plastic damage constitutive model [16], a micromechanics-based constitutive model [17] or an energy-based constitutive model [18] cannot be defined using USDFLD. UMAT (User subroutine to define a material’s mechanical behavior) has several advantages over USDFLD, including the modification of constitutive relations and consideration of uncertainty in material properties [19].

In most related researched, the maximum-stress criterion has often been used to predict the initial fiber compression failure of a composite. As reported by in the literature [20,21], fiber kinking plays a key role in composite compression failure when composite components are subject to compressive loads in the fiber’s direction. Silvestre Taveira Pinho proposed that fiber kinking was caused by shear-dominated matrix failure in a misaligned frame under significant longitudinal compression, and a set of LaRC05 failure criteria was proposed based on plasticity theory [22]. For matrix failure prediction, the effect of in-situ strength was considered using LaRC05 criteria based on Puck failure criteria, which use a potential fracture plane parallel to the fiber direction to describe the failure of matrices based on the Mohr–Coulomb theory [23,24]. Although the phenomenological LaRC05 criteria and Puck criteria possess high accuracy and received high recognition at the Second World-Wide Failure Exercise, they are not as widely used in the study of progressive damage of composite bolted joint bearing problems as the classical Hashin and maximum-stress criteria. The main reason for this may be that it is computationally expensive to calculate fiber misalignment angles and fracture angles of matrices especially in implicit finite elements due to convergence problems.

In this paper, an efficient method of determining the maximum fiber misalignment angle and maximum fracture angle of matrices is presented based on the derivatives of continuous functioning and using LaRC05 and Puck failure criteria. To improve the accuracy of numerical simulation, LaRC05, Puck and Hashin criteria (which were ranked very highly at the Second World-Wide Failure Exercise), as well as non-phenomenal maximum-stress criteria, were combined with different damage variables and implemented in the ABAQUS subroutine UMAT. A static tensile experiment was carried out on composite bolted joints to compare with simulation results, and not only were the numerical differences considered, but also the damage areas of composite plates and the deformations of fasteners.

## 2. Experiment Procedure

### 2.1. Description of Specimen

The single lap joint is a common means of joining airframe components on plate structures. In the experimental study, single-lap composite bolted joints were tested that consisted of IMS-977-2 carbon fiber/epoxy matrix composite lap plates joined by HST12 Hi-Lite fasteners. The geometry and size of specimens is shown in Figure 1, following the ASTM D5961 standard [25]. The stacking sequence of the laminated plate was [45°/90°/–45°/0°/90°/0°/–45°/90°/45°/–45°]s, for a total of 20 layers with single-layer nominal thicknesses of 0.188 mm. The HST12 Hi-Lite fastener with self-locking characteristics consisted of titanium alloy Ti–6Al–4V pins and stainless steel CRES347 nuts.

### 2.2. Experimental Process

Experimental tests were carried out at room temperature (24 ± 3 °C) and humidity (55 ± 5%), and the specimen rested in these room/laboratory conditions for three hours. As shown in Figure 2, the tensile experiment was performed on a CMT5504 Electronic universal testing machine (MTS System Corporation, Eden Prairie, MN, USA) with a 100-kN load capability. The relative error of force and displacement indication of the machine was ±0.5%. The grip holder moving speed was adjusted by step less speed regulation, and the accuracy of moving speed could be controlled within ±0.5%. Each specimen contained two support plates bonded to composite components to minimize the eccentricity in applied force from the grip holder of the test machine, and the thickness of support plates was equal to the sum of the composite plate thicknesses.

After the machine was preheated for more than 15 min, the grip holder was pulled at a speed of 2 mm/min to simulate quasi-static loading, then stopped after the load dropped by approximately 30% from the maximum value. The numerical values of applied load and grip holder displacement were recorded automatically. The test procedure corresponds to the guidelines given in ASTM D5961 [25].

## 3. Finite Element Model

### 3.1. Material Properties

The basic mechanical property parameters of the IMS-977-2 composite lamina are given in Table 1 [26,27]. The stress-strain curves of titanium alloy Ti–6Al–4V and stainless steel CRES347 are shown in Figure 3 [28], and the corresponding relationship between stress and strain (Table 2) can be obtained according to the stress-strain curves.

### 3.2. Progressive Damage Model

Many failure criteria have been proposed to predict fiber reinforced composite material failure. The early failure criteria did not consider the actual failure mechanisms, whereas later phenomenological failure criteria considered how failure would occur as long as the stress reached its ultimate strength (i.e., maximum-stress criteria), as shown in Equations (1)–(4). Equations (5)–(8) depict four Hashin criteria failure modes that are considered to be the earliest three-dimensional stress states and failure mechanisms. Micromechanical behavior was considered according to Puck criteria, where the degree of fiber failure is not only related to the stress state of the composite but also to the volume fraction of the fiber and matrices, as shown in Equations (9)–(15). For matrix failure, Puck thought that the matrix fracture occurred on the plane parallel to the fiber direction, while the fracture angle was used to describe the deflection of the fracture plane. The range of the fracture angle was from 0° to 180°, meeting the requirement of the maximum damage factor. In LaRc05 criteria, as shown in Equations (16)–(26), Pinho introduced the effects of in-situ strength to predict matrix criteria based on Puck criteria, and as a result fiber kinking prediction was built based on plasticity theory, which is different from the current damage criteria.

To search for the fracture angle of a matrix, Puck proposed the solution of a stepwise calculation of the maximum damage factor using an interval of 1° within its limits. For each stress state, 180 calculations are required, which is the same problem as for LaRc05 criteria when searching for the fiber misalignment angle. This may be a computationally expensive calculation, and severe convergence problems may occur in implicit finite element. For these reasons, a certain method is proposed in this paper based on the derivatives of continuous functions. As shown in Figure 4, several angles were selected that could derive a damage factor equal to zero, and the damage factors corresponding to these angles were calculated. The maximum damage factor and required matrix fracture angle or fiber misalignment angle were selected more quickly than the conventional method.

$\tau_{T}$ and $\tau_{L}$ are the transverse shear stress and longitudinal shear stress on the potential fracture plane, respectively; $\sigma_{N}$ is the normal stress on the potential fracture plane; α is fracture angle; $S_{T}$ and $S_{L}$ are transverse fracture resistance and longitudinal resistance on the potential fracture plane, respectively; $\mu_{T}$ and $\mu_{L}$ are the inclination coefficient or frictional coefficient, respectively, which represent the influence of normal stress on the fracture resistance [22].

The angle of the kink band, ψ, is found numerically in the range 0° and 180° so as to maximize the failure index in Equation (17). The misalignment angle, φ, is the sum of the initial misalignment angle φ0 (manufacturing defect) and the shear strain expressed in a coordinate system aligned with the manufacturing defect. This is calculated based on the linear shear response assumption. The McCauley brackets are defined as <x>+ = max {0,x}.

The above failure criteria in Table 3 were combined with the exponential damage variable, as shown in Equation (27) and sudden drop damage variable as shown in Equations (28) and (29) together, and implemented in the ABAQUS user-defined material subroutine UMAT to study the effect of progressive damage on the numerical simulation results.

The exponential damage variable is written as
(27)$$d=1-\frac{e^{\left(-T\varepsilon^{t}L\left(F-1\right)/G\right)}}{F}$$
where *F* is the value of the failure criterion, *T* is the coefficient in the stiffness matrix (e.g., $C_{11}$, $C_{22}$, $C_{33}$), $\varepsilon^{t}$ is ultimate failure strain, *L* is element characteristic length, and *G* is fracture toughness [29].

The sudden drop damage variable is written as
(28)d=0 (F<1)
(29)d=1 (F≥1)

## 4. Discussion

### 4.1. The Effect of Failure Criteria

The load-displacement curves obtained from the experiment and finite element method are presented in Figure 5 as different failure criteria but using the same exponential damage variables. The ultimate failure loads calculated by the finite element method were close to the experimental results. The maximum difference between the numerical simulation and the average value (17612N) of the experiment test was 4.8%, as calculated by the maximum-stress and Hashin criteria. The minimum difference was 1.1%, as calculated by the LaRC05 criteria. On the other hand, the difference in joint stiffness in the linear elastic stage was relatively larger (the maximum and minimum difference were 29.4% and 17.7%, respectively), and the curves obtained by the finite element method did not have a marked decline around the end points when compared with the experiment results. Possible reasons for these differences are as follows: (1) to achieve a good convergence when simulating the failure of a composite in ABAQUS, the viscous regularization of damage variables is often introduced to delay the degradation of the composite; (2) the material properties of fasteners are defined as elastic-plastic, while the protruding heads of fasteners fractured during the experimental tests, as shown in Figure 6.

The microscope images of damaged composite plates in the experimental tests and the prediction of composite failure in the finite element are presented in Table 4. The bolt hole of the composite plate was squeezed seriously by the bolt shank, causing several damage modes to occur such as matrix crushing, fibers pulling out and fiber fracture. It can be seen that while the maximum stress, Hashin and Puck criteria presented the same accuracy of prediction for fiber compression, the LaRc05 criteria could predict the boundary of the damaged region well. For the matrix failure prediction, Larc05 and Puck criteria were able to predict the damaged area of the matrix accurately; however, the damaged area predicted by Hashin criteria was obviously larger than the experimental result, while the damaged area was much smaller when predicted using maximum-stress criteria.

### 4.2. The Effect of Damage Variables

When the same failure criteria (LaRC05) were introduced with different damage variables in UMAT, the numerical simulation results showed an obvious difference. Compared to the use of exponential damage variables, sudden decrease damage variables reduced the bearing strength by 2.1%, as shown in Figure 7. When the sudden decrease damage variables were used, the load-displacement curve began to show a significant downward trend after the ultimate failure load was reached, which is close to the experimental results. Unlike sudden decrease damage variables, which are directly equal to 1 when the value of the failure criteria expression is >1, exponential damage variables increase from zero to 1 gradually with increases in the value of the criterion expression. The matrix stiffness of the composite presented faster degradation when the sudden decrease damage variable was used, and the damage propagation on the composite plates was faster as well.

Moreover, there was plastic deformation on the bolt head, screw and nut when the sudden decrease damage variable was used in UMAT, which is closer to the experimental phenomenon shown in Figure 8b,c. When the exponential damage variable was used, the plastic deformation of the fastener occurred only in the middle of the screw, as shown in Figure 8a.

### 4.3. The Effect of Subroutine

As the same failure criteria, damage variables and damage stiffness matrix were used in subroutine USDFLD and UMAT, respectively, the difference in the ultimate failure load was 14.2%, as shown in Figure 9. There were severe non-convergence in the USDFLD calculation process when the material property of the bolt was defined as elastic-plastic, but this was avoided in the UMAT calculation process. Unlike subroutine USDFLD, which can call strain and stress directly, subroutine UMAT can only call strains directly, then calculate stress according to the damage stiffness matrix and strain of the composite. The damage variable is the regularized viscosity before the calculation of stress, making the damage variable smaller than the real value and always less than 1, as shown in Equation (30). In other words, the degradation of composite stiffness was delayed by the viscous regularization of the damage variable, improving the convergence of the calculation.
(30)$$d^{V}=\frac{\eta }{\eta +\Delta t}*d^{\prime }+\frac{\Delta t}{\eta +\Delta t}*d^{\prime }$$
where $d^{V}$ is damage variable of the current incremental step with regularized viscosity, η is the viscosity coefficient, Δt is the time increment, and $d^{\prime }$ is the damage variable of the previous incremental step with regularized viscosity.

## 5. Conclusions

In order to improve the accuracy and efficiency of simulating the bearing behaviors of composite bolted joints, the effects of failure criteria, damage variables and user subroutines were studied in this paper, and experimental tests were carried out to compare numerical results. An approach based on a derivative method was used to find the fiber misalignment angle and the matrix fracture angle by applying LaRc05 criteria, and it was found this method had better efficiency than the conventional ergodic method.

When combined with the same damage variable, the maximum stress, Hashin and Puck criteria all presented the same accuracy at predicting fiber compression, and the LaRc05 criteria were able to predict the boundary of the damaged region well. LaRc05 and Puck criteria presented more accurate results than maximum-stress and Hashin criteria at predicting the matrix failure.

Compared to exponential damage variables, using the sudden decrease damage variables in UMAT could more accurately replicate experimental results when combined with LaRc05 criteria. As the same criteria and damage variables were incorporated in different user subroutines, there was better convergence in UMAT calculation than USDFLD calculation.

## Figures and Tables

**Figure 1 materials-13-05606-f001:**
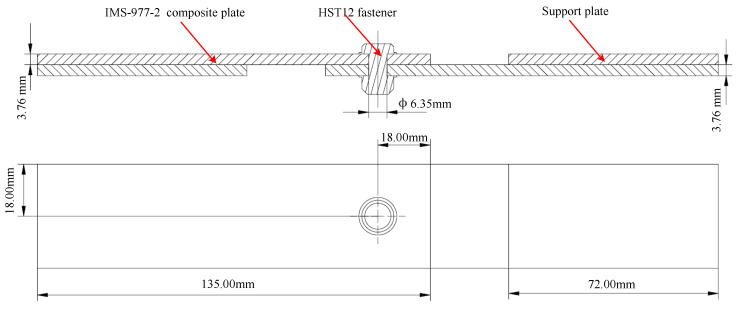
Specimen geometry size.

**Figure 2 materials-13-05606-f002:**
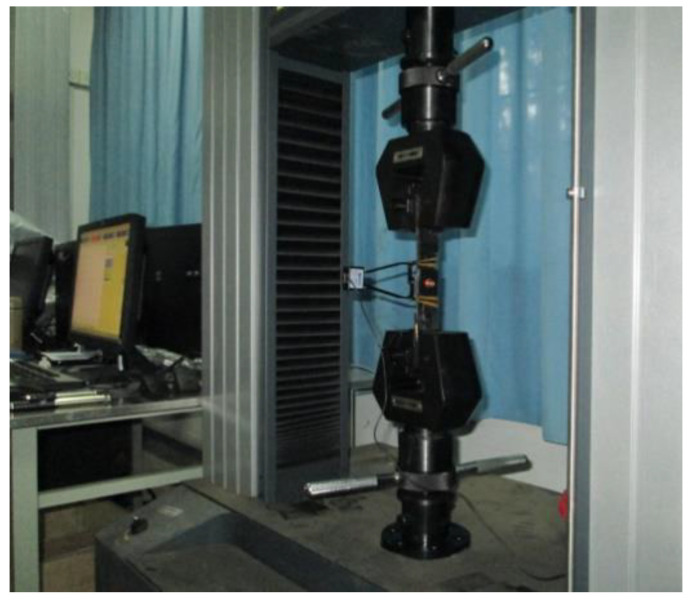
Tensile experiment.

**Figure 3 materials-13-05606-f003:**
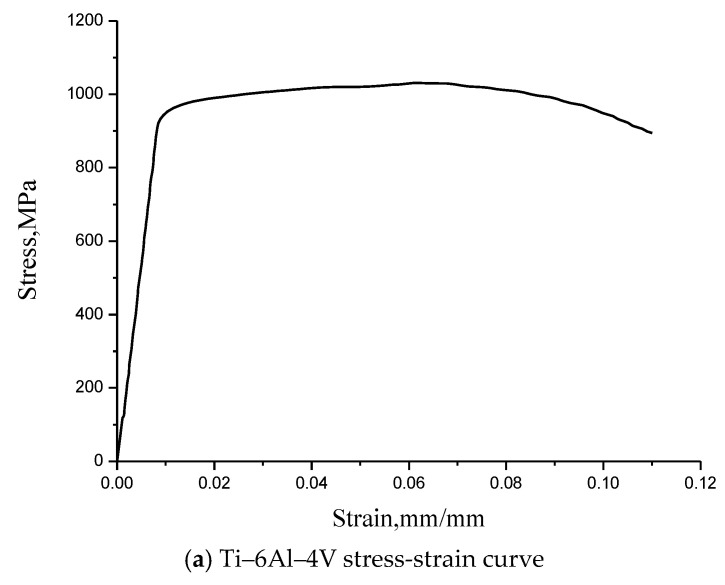
Stress-strain curves of fastener.

**Figure 4 materials-13-05606-f004:**
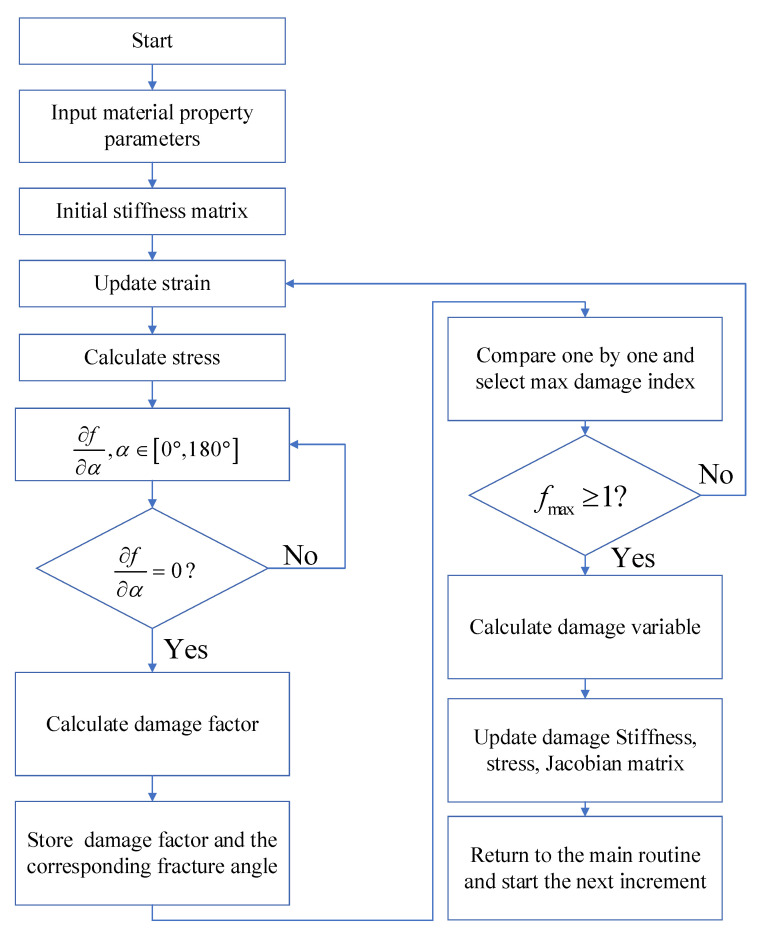
Flow chart of the search for fiber misalignment and the matrix fracture angle.

**Figure 5 materials-13-05606-f005:**
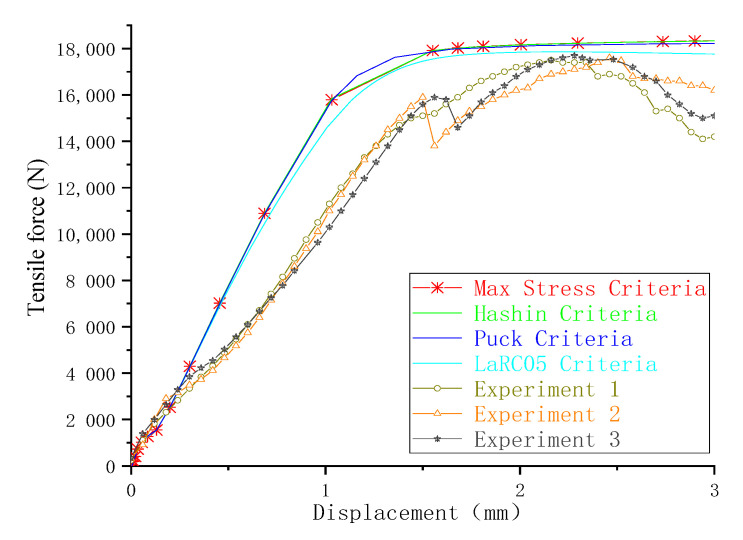
Force-displacement curves.

**Figure 6 materials-13-05606-f006:**
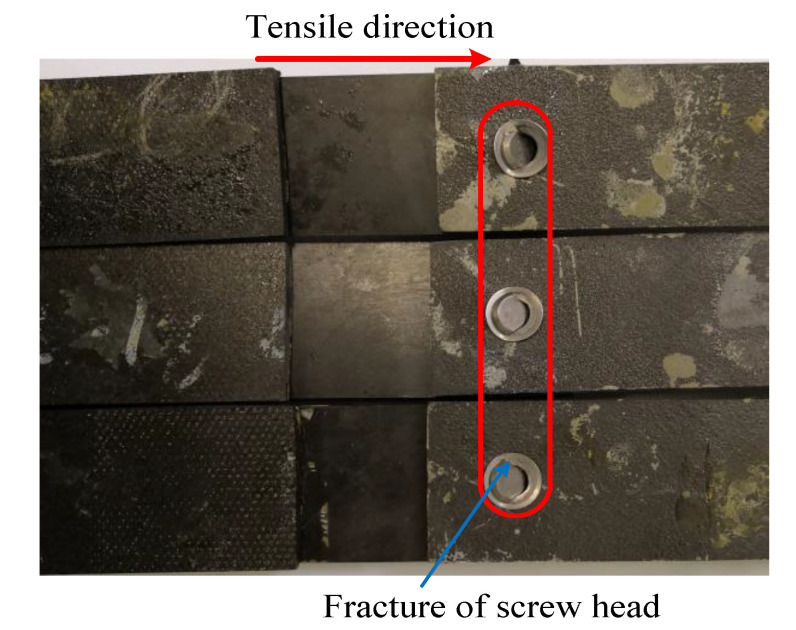
Experiment specimens.

**Figure 7 materials-13-05606-f007:**
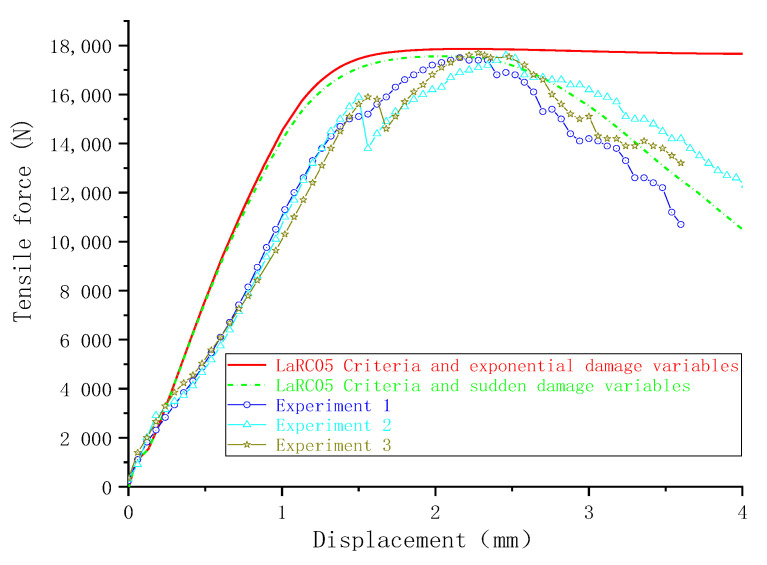
Force-displacement curves obtained with different damage variables.

**Figure 8 materials-13-05606-f008:**
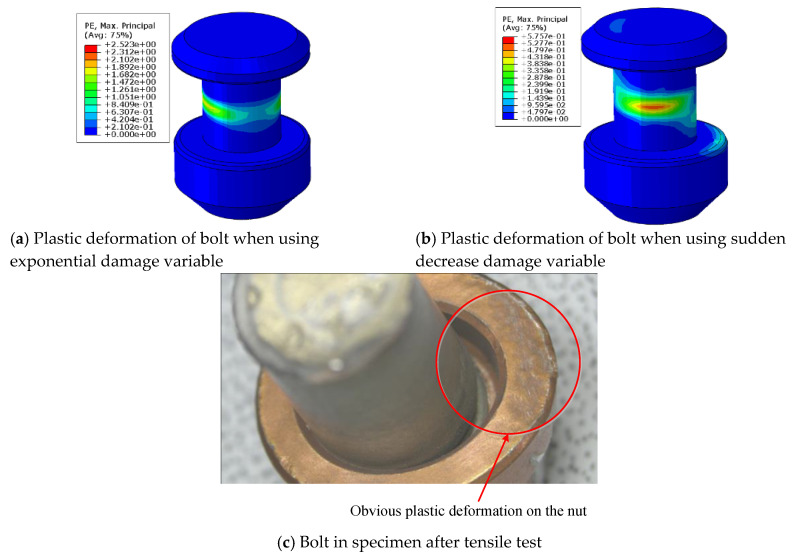
Plastic deformation of a bolt.

**Figure 9 materials-13-05606-f009:**
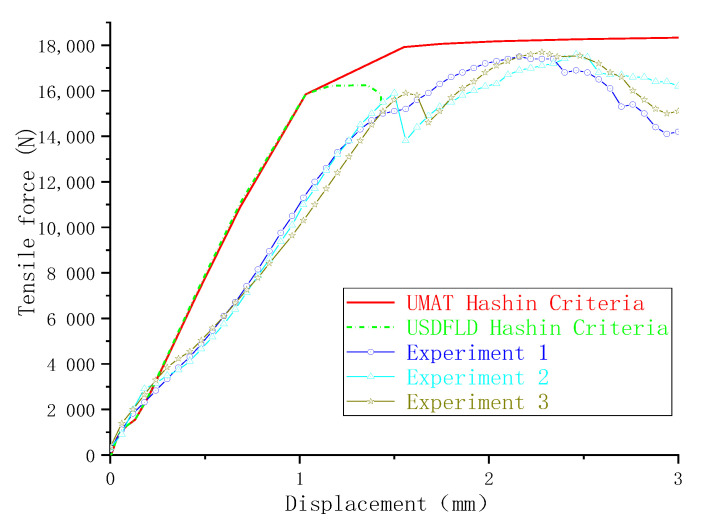
Force-displacement curves obtained with different subroutines.

**Table 1 materials-13-05606-t001:** Mechanical parameters of the lamina.

Property	Units	Value
Longitudinal tensile modulus, E_11_	GPa	156.00
Transverse tensile modulus, E_22_, E_33_	GPa	8.35
In-plane shear modulus, G_12_	GPa	4.20
Out-of-plane shear modulus, G_13_	GPa	4.20
Out-of-plane shear modulus, G_23_	GPa	2.52
Major Poisson’s ratio, $\upsilon_{12}$	-	0.33
Through thickness Poisson’s ratio, $\upsilon_{13}$	-	0.33
Through thickness Poisson’s ratio, $\upsilon_{23}$	-	0.55
Longitudinal tensile strength, X_T_	MPa	2500.00
Longitudinal compressive strength, X_C_	MPa	1400.00
Transverse tensile strength, Y_T_	MPa	75.00
Transverse compressive strength, Y_C_	MPa	250.00
In-plane shear strength, S_12_	MPa	95.00
In-plane shear strength, S_13_	MPa	95.00
Out-of-plane shear strength, S_23_	MPa	108.00
Fiber tensile fracture energy, G_ft_	J/m^2^	91.60
Fiber compressive fracture energy, G_fc_	J/m^2^	79.90
Matrix tensile fracture energy, G_mt_	J/m^2^	0.22
Matrix compressive fracture energy, G_mc_	J/m^2^	2.00

**Table 2 materials-13-05606-t002:** Material properties of the HST (Hi-Lite fastening system) fastener.

Specimen	Modulus	Poisson Ratio
Hi-Lite pins	1.103 × 10^2^ GPa	3.100 × 10^−1^
Hi-Lite nuts	2.110 × 10 ^2^ Gpa	2.790 × 10^−1^
	$\sigma_{y}$	$\varepsilon_{p}$
Stress $\sigma_{y}$ (Mpa) and strain $\varepsilon_{p}$ of Ti–6Al–4V in the plastic deformation stage	8.560 × 10^2^	0
	9.380 × 10^2^	8.870 × 10^−^^3^
	9.740 × 10^2^	1.300 × 10^−^^2^
	9.820 × 10^2^	1.580 × 10^−^^2^
	9.910 × 10^2^	2.010 × 10^−^^2^
	1.010 × 10^3^	3.750 × 10^−^^2^
	1.020 × 10^3^	5.070 × 10^−^^2^
	1.030 × 10^3^	6.790 × 10^−2^
Stress $\sigma_{y}$ (Mpa) and strain $\varepsilon_{p}$ of CRES347 in the plastic deformation stage	3.940 × 10^2^	0
	6.370 × 10^2^	3.180 × 10^−3^
	6.450 × 10^2^	3.240 × 10^−3^
	7.270 × 10^2^	4.050 × 10^−3^
	7.520 × 10^2^	4.470 × 10^−3^
	7.860 × 10^2^	5.470 × 10^−3^
	8.000 × 10^2^	6.520 × 10^−3^
	8.020 × 10^2^	6.710 × 10^−3^
	8.040 × 10^2^	7.480 × 10^−3^

**Table 3 materials-13-05606-t003:** Failure criteria.

Maximum-stress criteria	Fiber Tensile Failure	
	${\left(\frac{\sigma_{11}}{X_{T}}\right)}^{2}\geq 1\;\left(\sigma_{11}>0\right)$	(1)
	Fiber Compressive Failure	
	${\left(\frac{\sigma_{11}}{X_{C}}\right)}^{2}\geq 1\;(\sigma_{11}<0)$	(2)
	Matrix Tensile Failure	
	${\left(\frac{\sigma_{22}}{Y_{T}}\right)}^{2}\geq 1\;\left(\sigma_{22}>0\right)$	(3)
	Matrix Compressive Failure	
	${\left(\frac{\sigma_{22}}{Y_{C}}\right)}^{2}\geq 1\;\left(\sigma_{22}<0\right)$	(4)
Hashin criteria	Fiber Tensile Failure	
	${\left(\frac{\sigma_{11}}{X_{T}}\right)}^{2}+{\left(\frac{\tau_{12}}{S_{12}}\right)}^{2}+{\left(\frac{\tau_{13}}{S_{13}}\right)}^{2}\geq 1\;\left(\sigma_{11}>0\right)$	(5)
	Fiber Compressive Failure	
	${\left(\frac{\sigma_{11}}{X_{C}}\right)}^{2}\geq 1\;(\sigma_{11}<0)$	(6)
	Matrix Tensile Failure	
	${\left(\frac{\sigma_{22}+\sigma_{33}}{Y_{T}}\right)}^{2}+\frac{1}{{\left(S_{23}\right)}^{2}}\left(\tau_{23}{}^{2}-\sigma_{22}\sigma_{33}\right)+{\left(\frac{\tau_{12}}{S_{12}}\right)}^{2}+{\left(\frac{\tau_{13}}{S_{13}}\right)}^{2}\geq 1\;$ $\left(\sigma_{22}+\sigma_{33}>0\right)$	(7)
	Matrix Compressive Failure	
	$\frac{\sigma_{22}+\sigma_{33}}{Y_{C}}\left[{\left(\frac{Y_{C}}{2S_{23}}\right)}^{2}-1\right]+{\left(\frac{\sigma_{22}+\sigma_{33}}{2S_{23}}\right)}^{2}+\frac{\tau_{23}{}^{2}-\sigma_{22}\sigma_{33}}{{\left(S_{23}\right)}^{2}}+{\left(\frac{\tau_{12}}{S_{12}}\right)}^{2}+{\left(\frac{\tau_{13}}{S_{13}}\right)}^{2}\geq 1\;\left(\sigma_{22}+\sigma_{33}<0\right)$	(8)
Puck criteria	Fiber Tensile Failure	
	$\frac{1}{X_{T}}\left(\sigma_{11}-\left(\nu_{12}-\nu_{12}^{f}\cdot m\frac{E_{11}}{E_{11}^{f}}\right)\left(\sigma_{22}+\sigma_{33}\right)\right)\geq 1\;\left(\sigma_{11}\geq 0\right)$	(9)
	Fiber Compressive Failure	
	$\frac{1}{X_{C}}\left(\sigma_{11}-\left(\nu_{12}-\nu_{12}^{f}\cdot m\frac{E_{11}}{E_{11}^{f}}\right)\left(\sigma_{22}+\sigma_{33}\right)\right)\geq 1\;\left(\sigma_{11}<0\right)$	(10)
	Matrix Tensile Failure	
	${\left(\frac{\sigma_{N}}{Y_{T}}\right)}^{2}+{\left(\frac{\tau_{T}}{S_{T}}\right)}^{2}+{\left(\frac{\tau_{L}}{S_{L}}\right)}^{2}\geq 1\;\left(\sigma_{N}>0\right)$	(11)
	Matrix Compressive Failure	
	${\left(\frac{\tau_{T}}{S_{T}-\mu_{T}\sigma_{N}}\right)}^{2}+{\left(\frac{\tau_{L}}{S_{L}-\mu_{L}\sigma_{N}}\right)}^{2}\geq 1\;\left(\sigma_{N}\leq 0\right)$	(12)
	$\sigma_{N}=\frac{\sigma_{2}+\sigma_{3}}{2}+\frac{\sigma_{2}-\sigma_{3}}{2}cos\left(2\alpha \right)+\tau_{23}sin\left(2\alpha \right)$	(13)
	$\tau_{T}=-\frac{\sigma_{2}-\sigma_{3}}{2}sin\left(2\alpha \right)+\tau_{23}cos\left(2\alpha \right)$	(14)
	$\tau_{L}=\tau_{12}cos\left(\alpha \right)+\tau_{31}sin\left(\alpha \right)$	(15)
LaRc05 Criteria	Fiber Tensile Failure	
	$\frac{\langle \sigma_{11}\rangle {}_{+}}{X_{T}}\geq 1\;\left(\sigma_{11}>0\right)$	(16)
	Fiber Kinking Failure	
	${\left(\frac{\tau_{23}^{m}}{S_{T}^{is}-\eta_{T}\sigma_{2}^{m}}\right)}^{2}+{\left(\frac{\tau_{12}^{m}}{S_{L}^{is}-\eta_{L}\sigma_{2}^{m}}\right)}^{2}+{\left(\frac{\langle \sigma_{2}^{m}\rangle {}_{+}}{Y_{T}^{is}}\right)}^{2}\geq 1\;(\sigma_{11}<0)$	(17)
	Matrix Tensile Failure	
	${\left(\frac{\tau_{23}^{m}}{S_{T}^{is}}\right)}^{2}+{\left(\frac{\tau_{23}^{m}}{S_{T}^{is}}\right)}^{2}+{\left(\frac{\langle \sigma_{2}^{m}\rangle {}_{+}}{Y_{T}^{is}}\right)}^{2}\geq 1\;\left(\sigma_{N}\geq 0\right)$	(18)
	Matrix Compressive Failure	
	${\left(\frac{\tau_{23}^{m}}{S_{T}^{is}-\eta_{T}\sigma_{2}^{m}}\right)}^{2}+{\left(\frac{\tau_{23}^{m}}{S_{T}^{is}-\eta_{L}\sigma_{2}^{m}}\right)}^{2}+{\left(\frac{\langle \sigma_{2}^{m}\rangle {}_{+}}{Y_{T}^{is}}\right)}^{2}\geq 1\;\left(\sigma_{N}<0\right)$	(19)
	$\sigma_{2}^{\psi }=cos^{2}\psi \sigma_{2}+sin^{2}\psi \sigma_{3}+2sin\psi cos\psi \tau_{23}$	(20)
	$\tau_{12}^{\psi }=\tau_{12}cos\psi +\tau_{31}sin\psi$	(21)
	$\tau_{23}^{\psi }=-sin\psi cos\psi \sigma_{2}+sin\psi cos\psi \sigma_{3}+\left(cos^{2}\psi -sin^{2}\psi \right)\tau_{23}$	(22)
	$\tau_{31}^{\psi }=\tau_{31}cos\psi -\tau_{12}sin\psi$	(23)
	$\sigma_{2}^{m}=sin^{2}\varphi \sigma_{1}+cos^{2}\varphi \sigma_{2}^{\psi }-2sin\varphi cos\varphi \tau_{12}^{\psi }$	(24)
	$\tau_{12}^{m}=-sin\varphi cos\varphi \sigma_{1}+sin\varphi cos\varphi \sigma_{2}^{\psi }+\left(cos^{2}\varphi -sin^{2}\varphi \right)\tau_{12}^{\psi }$	(25)
	$\tau_{23}^{m}=\tau_{23}^{\psi }cos\varphi -\tau_{31}^{\psi }sin\varphi$	(26)

**Table 4 materials-13-05606-t004:** Failure phenomena of composite plates.

Failure of composite plate in experimental test	Failure of composite plate in finite element simulation (SDV2 is the degree of fiber compression or kinking failure; SDV4 is the degree of matrix compression failure)
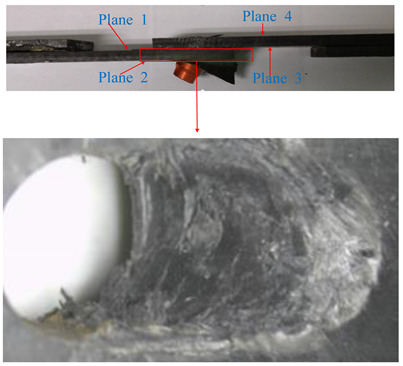 (a) Viewing above the Plane 1	**(1) Maximum Stress** 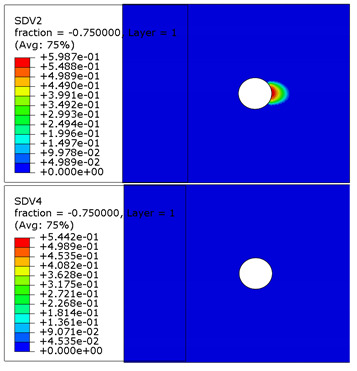 **(2) Hashin** 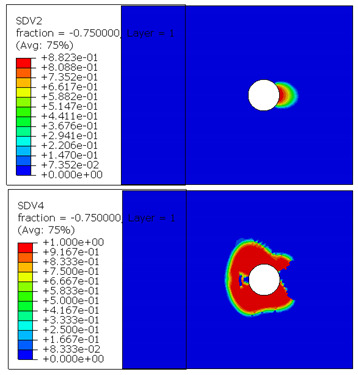 **(3) Puck** 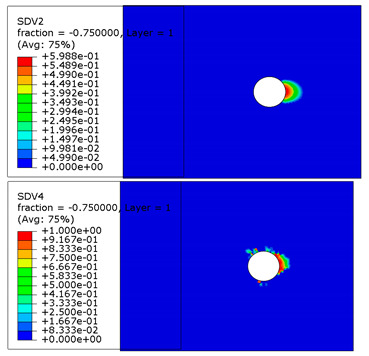 **(4) LaRC05** 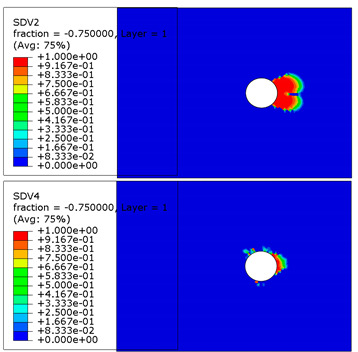
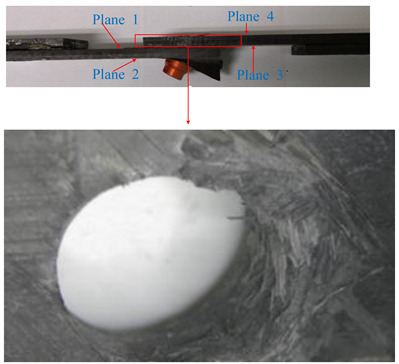 (b) Viewing above the Plane 3	**(5) Maximum Stress** 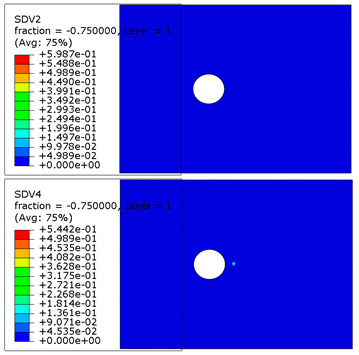 **(6) Hashin** 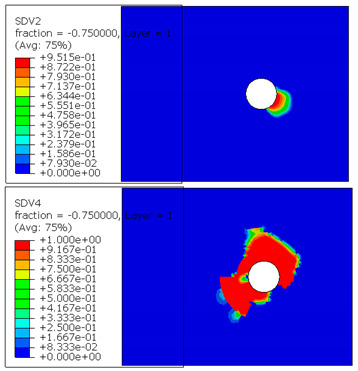 **(7) Puck** 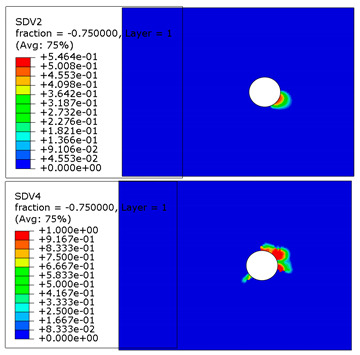 **(8) LaRC05** 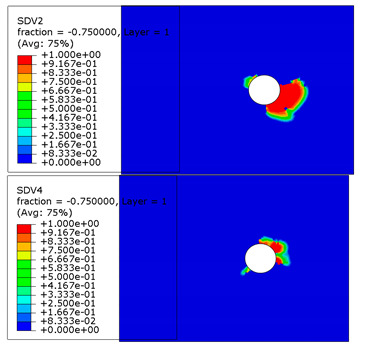

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
