# Peer review of "Progressive Damage Numerical Modelling and Simulation of Aircraft Composite Bolted Joints Bearing Response"

_materials, 2020, doi:10.3390/ma13245606_

Round 1

Reviewer 1 Report

Dear authors.

In my opinion, your paper present good results to improve the simulation of composite bolted joints. It means that it can be considered acceptable for publication, nevertheless, some changes must be done.

Reviewer 2 Report

Authors use abbr. which are familiar within particular filed of expertise, but not commonly. Some terms should be referenced (e.g. Second World-Wide Failure Exercise).

Most figures should be improved, readability is low

Fig.5, Max Stress Criteria (red color curve) is missing

Tables 1, 2 should not be broken, table 2 is confusing, fig 3b and data on table 2 should be checked, there should be mentioned if the Engineering or True Stress stress-strain curve is used further. 

The equation should be correctly typed, not inserted as a picture, conditions should  separated
